# Synovial-Fluid miRNA Signature for Diagnosis of Juvenile Idiopathic Arthritis

**DOI:** 10.3390/cells8121521

**Published:** 2019-11-26

**Authors:** Nadége Nziza, Eric Jeziorski, Marion Delpont, Maïlys Cren, Hugues Chevassus, Aurélia Carbasse, Perrine Mahe, Hamouda Abassi, Pauline Joly-Monrigal, Eric Schordan, Alain Mangé, Christian Jorgensen, Florence Apparailly, Isabelle Duroux-Richard

**Affiliations:** 1IRMB, INSERM, Univ. Montpellier, F-34000 Montpellier, France; 2Arthritis R&D, F-92200 Neuilly sur Seine, France; 3Pediatric Department, CHU Montpellier, Univ. Montpellier, F-34000 Montpellier, France; 4Pathogenesis and Control of Chronic Infections, CHU Montpellier, Univ. Montpellier, INSERM, F-34000 Montpellier, France; 5Pediatric Orthopedic Surgery Unit, CHU Montpellier, Univ. Montpellier, F-34000 Montpellier, France; 6Centre d’Investigation Clinique, CHU Montpellier, F-34000 Montpellier, France; 7Centre d’Investigation Clinique 411, INSERM, F-34000 Montpellier, France; 8FIRALIS SA, F-68330 Huningue, France; 9IRCM, INSERM, Univ. Montpellier, ICM, F-34000 Montpellier, France; 10Clinical Department for Osteoarticular Diseases, CHU Montpellier, Univ. Montpellier, F-34000 Montpellier, France

**Keywords:** miRNA, juvenile arthritis, biomarker, synovial fluid

## Abstract

Juvenile idiopathic arthritis (JIA) is the most common chronic inflammatory rheumatism in childhood; microRNAs (miRNAs) have been proposed as diagnostic biomarkers. Although joints are the primary targets for JIA, a synovial fluid-based miRNA signature has never been studied. We aim to identify miRNA biomarkers in JIA by comparing synovial fluid and serum samples from children with JIA and *K. kingae* septic arthritis (SA). With next-generation high-throughput sequencing, we measured the absolute levels of 2083 miRNAs in synovial fluid and serum from an exploratory cohort of children and validated differentially expressed miRNAs in a replication study by using RT-qPCR. We identified a 19-miRNA signature only in synovial fluid samples that was significantly deregulated, with at least 2-fold change in expression, in JIA versus SA (*p* < 0.01). The combination of miR-6764-5p, miR-155, and miR-146a-5p expression in synovial fluid yielded an area under the receiver operating characteristic curve of 1 (95% CI 0.978 to 1), thereby perfectly differentiating JIA from SA in children. We propose, for the first time, a synovial fluid-specific miRNA signature for JIA and associated signaling pathways that may indicate potential biomarkers to assist in the classification and differential diagnosis of JIA and help in understanding JIA pathogenesis.

## 1. Introduction

Juvenile idiopathic arthritis (JIA) is the most common chronic inflammatory disease in childhood and is characterized by arthritis and systemic features. JIA represents a heterogeneous group of inflammatory arthropathies, whose classification has been recently revised by the International League of Associations for Rheumatology (ILAR) [1]. Historically, its management and classification have been based on adult rheumatisms. Thus, treatment options for JIA are derived in large part from protocols used in adults and include non-steroidal anti-inflammatory drugs, intra-articular injections of glucocorticoids, disease-modifying anti-rheumatic drugs (DMARDS), and biological therapeutics such as tumor necrosis factor inhibitors [2,3].

However, although childhood and adult rheumatic diseases share some similarities, they have fundamental differences such as clinical features, prognosis, and pathophysiology, which are not yet well understood. For example, rheumatoid arthritis (RA) is clearly an autoimmune disease strongly associated with autoantibodies against citrullinated peptides and IgG (rheumatoid factor [RF]), which are used for diagnosis and are involved in the disease pathophysiology [4]. However, the most frequent form of JIA, oligoarticular JIA, does not have a strong association with autoimmunity; the presence of anti-nuclear antibodies (ANAs) is inconsistent, and the implication of ANAs in the JIA pathophysiology is not clearly understood. The new ILAR classification now considers that only systemic JIA, RF-positive JIA, and enthesitis/spondylitis-related JIA are the juvenile counterparts of diseases also observed in adults, with other JIA diseases occurring only in children (early-onset ANA-positive JIA). In this context, identifying specific biomarkers is essential to help characterize and classify JIA diseases for early diagnosis and proposing molecules related to specific pathophysiological pathways for future therapeutic innovation.

Because of their specificity and stability as well as their detectable presence in body fluids, microRNAs (miRNAs) have been proposed as diagnostic biomarkers for many diseases. They are short non-coding RNAs involved in the regulation of multiple biological pathways by post-transcriptionally repressing the expression of protein-encoding genes. They have key functions in the regulation of almost every cellular process investigated so far. The altered expression of miRNAs has been described in many diseases, including rheumatisms such as RA [5,6,7,8,9]. In JIA, the implication of miRNAs remains poorly explored. Only a few studies have recently identified miRNAs with abnormal expression in JIA; they used blood samples (serum or plasma mainly), focusing on candidate miRNAs previously associated with adult diseases, and compared data with healthy donor data [10,11,12,13,14,15].

The present study aims to identify miRNA-based biomarkers to better characterize JIA diseases by using three novel tracks: (1) next-generation high-throughput screening; (2) synovial fluid, the main tissue involved in JIA diseases; and (3) children with septic arthritis (SA) as controls because of the inability to collect synovial fluid from joints of healthy children and because it is the most frequent aetiology in children hospitalized for arthritis, just before JIA, albeit involving different treatment and prognoses. In addition, there is currently a lack of biomarkers to diagnose SA in children at initial presentation, especially when disease features are not typical to differentiate septic and inflammatory arthritis. However, early diagnosis is critical to rapidly initiate antibiotic treatment in children with SA because joint destruction can start 8 h after bacterial inoculation [16,17]. However, in about 60% of cases, isolation of the causative agent, the gold standard for SA diagnosis, remains inconclusive or impossible to perform. Conventional infectious biomarkers are also controversial when used for SA diagnosis. Serum levels of procalcitonin and C-reactive protein (CRP) are not reliable for childhood SA [18,19]. Morphology analysis and count of polynuclear neutrophils, such as the observation of cytoplasmic vacuolation and toxic granulation formation, have been proposed as early markers of infection, but their sensitivity is still debated and the availability of an expert cytologist in clinical routine is often lacking [20]. Hence, our study could also help reveal specific and easily detectable markers for SA in children. 

## 2. Materials and Methods

### 2.1. Collection and Preparation of Synovial Fluid (SF) and Serum (SE) Samples for miRNA Studies

Children presenting acute arthritis requiring joint aspiration were included if they were age 6 months to 15 years and met the diagnosis criteria for JIA or SA. ILAR criteria were used to define JIA. Children with systemic JIA or receiving steroids, biological therapy, or DMARDS in the month preceding the puncture, were excluded. We included samples from SA patients with a bacteriological identification validated in cultured SF according to national guidelines or by specific PCR for *Kingella kingae*. Participants underwent standard explorations (i.e., blood cell count, CRP measurement, SF direct examination, and bacteriological analysis) that were systematic for SA and on-demand for JIA. A direct examination consisted of cell identification and count without cytological analysis. 

We obtained SF and SE samples from patients with JIA (n = 13) and SA (n = 13). SE samples were collected in BD vacutainer tubes (BD Diagnostics, Le Pont-de-Claix, France) according to standard procedures, then centrifuged at 1500× *g* for 10 min; whole blood was separated into serum and cellular fractions within 2 h after collection. SE samples were stored at −80°C. SF samples were collected according to clinical recommendations, and joint drainage was performed by experienced orthopaedic surgeons with the patient under local anesthesia and aseptic conditions. SF samples were collected in BD vacutainer tubes (BD Diagnostics, France), treated with heparin, centrifuged (300× *g*, 10 min) or not, aliquoted and stored at –80°C. 

We studied one exploratory cohort (JIA: n = 9; SA: n = 7) and a validation cohort (JIA: n = 9; SA: n = 9) (Appendix A). In the exploratory cohort, we included two different SF samples, with and without centrifugation, to evaluate the robustness of the identified miRNA signature (Appendix A). The study was approved by an ethics committee (Comité de Protection des Personnes sud méditerranée I; no. 214 R24). Informed consent was provided by the parents/guardians of the children in accordance with procedures approved by the local human ethics committee (2014-A01561-46).

### 2.2. miRNA Profiling

The biotechnology company FIRALIS SAS (Huningue, France) used a miRNA whole transcriptome assay (WTA) associated with next-generation sequencing for miRNome analysis, analyzing 15 µL body fluid. An miRNA profile of 2084 miRNAs (miRBase v20) was prepared by using the HTG EdgeSeq miRNA whole transcriptome targeted sequencing assay (HTG EdgeSeq, HTG Molecular Diagnostics, Tuscon, AZ, USA). This technology is based on a nuclease protection-targeted RNA sequencing assay that uses an extraction free lysis process followed by a nuclease protection assay to prepare a stoichiometric library of nuclease protection probes (NPPs) for measurement. 

#### 2.2.1. HTG EdgeSeq Analysis

Samples were prepared as follows: 15 µL plasma lysis buffer, 15 µL sample, and 3 µL proteinase K were mixed and incubated at 50 °C for 3 h with orbital shaking. An amount of 25 µL of the mixture was transferred to the HTG sample plate and loaded into the HTG processor for the nuclease protection assay and to prepare the stoichiometric NPP with the HTG EdgeSeq miRNA kit (HTG Molecular Diagnostics, Tucson, AZ, USA).

#### 2.2.2. Molecular Barcoding and Adapter Addition

For SF samples, barcoding involved using the Hemo KlenTaq enzyme (MO332S, NEB, Evry, France). For each sample, we mixed 2.4 µL Hemo KlenTaq, 0.6 µL dNTPs (10 nM) (NEB, N0447S), 6 µL OneTaq PCR GC Buffer 5X (B9023S, NEB), 3 µL forward and reverse primers (HTG EdgeSeq, HTG Molecular Diagnostics), 3 µL sample preparation, and 12 µL H_2_0. The PCR step involved using the ABI 2720 Thermocycler with the cycling profile 95 °C for 4 min followed by 16 cycles of 95 °C for 15 s, 55 °C for 45 s, and 68 °C for 45 s. The protocol ended with 68 °C for 10 min.

#### 2.2.3. PCR Clean-Up

Agentcour AMPure XP beads (A63880, Beckmancoulter, Villepinte, France) were used to remove excess primers from the library. For each sample, 37.5 µL AMPure XP beads were combined with 15 µL PCR product. After mixing 10 times by the use of a pipette, the solution was incubated for 5 min at room temperature and then placed on the magnetic stand. After 2 min to separate beads, the cleared solution was carefully removed without disturbing the beads. Beads were washed twice with 200 µL 80% ethanol. Elution of the PCR product linked to the beads involved using 25 µL H_2_0. The purified solution of the PCR product was placed in a new tube while the plate was on the magnetic stand to separate PCR products and beads.

#### 2.2.4. Determination of Library Concentration

To determine the library concentration for each sample, the Kapa Biosystems qPCR Kit (KK4824, Cliniscience, Nanterre, France) was used. For each reaction, the mixture consisted of 12 µL Mastermix, 0.4 µL ROX dye, 3.6 µL H_2_0, and 4 µL template (standards or library diluted at 1/10000). The samples were run on an ABI PRISM 7900HT system (High ROX) at 95 °C for 5 min, 35 cycles of 95 °C for 30 s, and 65 °C for 45 s, with data collection followed by a dissociation curve. Standards corresponded to an amplicon of 452 bp, and the amplicon of NPP with a barcode corresponded to 115 bp. The ratio was applied to determine the concentration of each library.

#### 2.2.5. Sample Pooling and Sequencing

Each sample was pooled to generate a pooled library at 4 nM. From this pooled library, 5 µL was mixed with 0.2 N freshly prepared NaOH and incubated for 2 min. The solution was vortexed briefly and centrifuged at 280× *g* for 1 min and mixed with 990 µL pre-chilled HT1 buffer (Illumina NextSeq Reagent v2 kit, Illumina, Paris, France). An amount of 5% PhiX (PhiX control v3, Illumina, Paris, France) at 20 pM was prepared, and 570 µL prepared denatured library at 20 pM was mixed with 30 µL 20 pM PhiX and loaded into the NextSeq High output v2 75 cycles kit and sequenced.

#### 2.2.6. Sequence Analysis

Sequencing data were first analyzed and checked by using the Q30 metric. Next-generation sequencing fulfilled Illumina recommendations. Data reconstruction and analysis were performed with FASTQ files from the Illumina NextSeq platform and processed by using HTG Parser software.

#### 2.2.7. Normalization

Before data normalization, negative control (ANT) quality control (QC) was performed on parsed raw data. When samples showed a high number of reads in negative control (>150 counts per million (CPM)), they were flagged as QC failures and removed from the analysis. The normalization involved 9 steps: (1) removal of the background of the sample (mean of the negative control), which was subtracted for all miRNAs; (2) all negative values set to 0; (3) data transformation in CPM for all samples; (4) logarithmic (base 10) transformation; (5) mean of each miRNA; (6) mean of the miRNAs subtracted for each miRNA; (7) re-transformation of the data with exponential function; (8) computation of the median of each sample; and (9) data before logarithm transformation (data of Step 3) divided by the median of each sample (data of Step 8).

### 2.3. miRNA Extraction and Quantification by RT-qPCR

Total RNA, including small RNA, was extracted from 100 μL SF by using the miRNeasy Serum/Plasma kit with a Qiacube (QIAGEN, Courtaboeuf, France) according to the manufacturer’s instructions. Reverse transcription of miRNAs and preamplification involved 2 μL RNA sample eluent with the TaqMan MicroRNA Reverse Transcription kit and TaqMan PreAmp Master Mix, respectively. Because of the small amount of SF, the TaqMan miRNA quantification method involved two preamplifications of the cDNA. Specificity, linearity, and efficacy of miRNA quantification was validated (Appendix A). Although EDTA tubes would have been preferable, SF samples were collected in heparinized tubes because of constraints related to the study. Nevertheless, because we compared samples from the same processing source in the present study, we were in accordance with MIQE guidelines for minimum information for the publication of RT-qPCR experiments [21]. The TaqMan reactions involved using TaqMan miRNA assays (ThermoFisher Scientific, Courtaboeuf, France) for the following miRNAs: hsa-miR-4417, hsa-miR-7150, hsa-miR-3687, hsa-miR-150-5p, hsa-miR-146a-5p, hsa-miR-6794-5p, hsa-miR-4800-5p, hsa-miR-4646-5p, hsa-miR-6782-5p, hsa-miR-4419a/b, hsa-miR-4667-5p, hsa-miR-155-5p, hsa-miR-339-3p, hsa-miR-342-5p, hsa-miR-6716-5p, hsa-miR-6734-3p, hsa-miR-6841-3p, hsa-miR-6764-5p, hsa-miR-8063, hsa-miR-2909, miR-648, and miR-4519. qPCR involved a ViiA 7TM system with a TaqMan fast advanced master mix (ThermoFisher Scientific). For each SE and SF sample, Ct values were normalized; mean Ct values were calculated and a normalization factor was applied based on this formula: normalization factor = 2 – (Mean Ct – sample Ct).

### 2.4. Statistical Analysis

The differentially expressed miRNA probe-sets were filtered with query parameters by using a signal threshold, the detection *p*-value, and the call rate to detect miRNAs (Perseus software) [22]. The false discovery rate (FDR) was determined with threshold 0.01 and log2 (fold change <–2 and > +2 or <–2 and >+4) to identify differences in miRNA expression. Heatmaps and hierarchical clustering involved using *z*-scores transformed from the original normalized values. GraphPad prism software was used for receiver operating characteristics (ROCs) and area under the ROC curve (AUC) determination. The generalized ROC criterion [23] with the best linear combination (virtual marker) of miRNA expression is the maximized AUC [24].

## 3. Results

To identify miRNAs differentially expressed in JIA versus SA, we analyzed the miRNome in SE and SF from the exploratory cohort (JIA: n = 5; SA: n = 3). Clinical and biological characteristics of participants are summarized in Table 1 and Appendix A. 

Of the 2083 miRNAs analyzed, >50% were detectable in SF and SE samples with CPM ≥20. To address differences in SE and SF miRNA levels between SA and JIA patients, data were analyzed by principal component analysis (PCA) and hierarchical clustering methods (Figure 1). 

PCA of all detectable miRNAs showed a separation between SA and JIA when quantified in SF samples (Figure 1A), which suggests specific disease-related changes in miRNA expression within inflamed joints. However, PCA of SE samples less clearly separated overall SA and JIA patterns. To identify miRNAs differently expressed between SA and JIA with statistical significance, we produced two different volcano plots for SF and SE (Appendix A). The threshold we used to screen up- or downregulated miRNAs was fold change in expression ≥2.0 and *p* < 0.05. Supervised hierarchical clustering showed that the miRNA signature in both body fluid samples distinguished the two groups of patients, albeit with a more discriminating profile in SF than SE samples (Figure 1B,C). As compared with 198 miRNAs in SE samples, 419 miRNAs in SF were differentially expressed between SA and JIA. When the *p*-value threshold was lowered (*p* < 0.01), 141 miRNAs were identified in SF (Appendix A) and none in SE samples.

### 3.1. A Signature Based on 21 miRNAs in SF Distinguishes JIA from SA

To reduce the list of 141 miRNAs differently expressed in SF from children with JIA and SA to reveal potential biomarkers useful in a clinical setting, we used new filter criteria. We established a list of miRNAs detected in only one of the two diseases with a threshold of at least 100 CPM and fold change in expression >4 or <–2 between JIA and SA. We identified 21 miRNAs differentially expressed in SF of children with JIA versus SA: 16 were upregulated and 5 downregulated (Table 2). 

Hierarchical clustering revealed perfect discrimination of the two disease groups (Figure 2A). Among the miRNAs downregulated in JIA versus SA, the most significantly downregulated were miR-6841-3p, miR-2909, and miR-8053, with CPM ≤50 in JIA samples, the detection limit for miRNAs when using RT-qPCR methods (Figure 2B). The most over-expressed miRNAs in JIA versus SA were miR-150-5p, miR-7150, and miR-4417, with CPM >1000 (Figure 2C).

### 3.2. An Independent Cohort Used to Validate the miRNA Signature Specific for JIA

To validate the clinical robustness of the candidate signature of 21 miRNAs differentially expressed in JIA versus SA, we used an HTG EdgeSeq analysis of an independent set of SF samples that were not centrifuged before storage (second exploratory cohort, see Appendix A). Hierarchical clustering revealed perfect discrimination between JIA and SA samples (Figure 2D) for the novel cohort, so differences in SF treatment before miRNA analysis did not affect the potential of our candidate miRNA signature (Table 2). Among the 21 miRNAs, 2 (miR-3687 and miR-4417) displayed a different expression profile in both exploratory and validation cohorts. Fold changes found in SF from Cohorts 1 and 2 were 4.42 and –1.13 for miR-3687 and 5.34 and 0.42 for miR-4417. In addition, with the recent update of the official miRBase database, miR-4417 is now removed and should not be considered as a miRNA. Overall, this replication study validated 19 of the 21 miRNAs as being specific to JIA, as compared with SA.

### 3.3. An SF Combination of Three miRNAs Differentiates JIA from SA at Initial Presentation

To further validate our results with another detection technique that is easy to perform in the clinic, we measured the expression of the 19 miRNAs identified in SF by using RT-qPCR analysis of a validation cohort (n = 9/group). Four SF miRNAs (miR-146a-5p, miR-150-5p, miR-155, and miR-342-5p) were significantly upregulated at *p* < 0.01 in JIA versus SA and 1 (miR-6764-5p) was significantly downregulated (Figure 3A).

Next, we used miRNA expression levels in SF as biological indicators for JIA and performed an ROC analysis to determine AUC values. AUC values were >0.9 for miR-155, miR-150-5p, miR-146a-5p, miR-6764-5p, and miR-342-5p, which suggests high diagnostic performance. To optimize the classification performance of miRNA levels in SF, we determined a new threshold by using mROC software. The combination of miR-6764-5p, miR-155, and miR-146a-5p in SF helped differentiate JIA from SA in children, with AUC = 1 (95% CI 0.978 to 1; Figure 3B). Overall, combining the detection of the three miRNAs in SF is an efficient biological indicator for JIA diagnosis, as compared with SA.

### 3.4. Comparative Pathway Analysis Reveals Key Pathophysiological Mechanisms

To examine which pathways were affected in JIA biological fluids, we used a miRWalk pathway analysis of the 19-miRNA signature identified. Overall, 19 WIKI biological processes were significantly enriched (*p* < 0.05, FDR-corrected), with at least two miRNAs involved in each pathway. The three prominent pathways were the insulin signaling pathway, involving 18 miRNAs of the signature (*p* = 3.4 × 10^−2^ to 1.3 × 10^−7^) and B-cell receptor and EGF/EFGR signaling pathways, involving 16 miRNAs (*p* = 4.8 × 10^−2^ to 6.5 × 10^−6^ and 4.5 × 10^−2^ to 2.4 × 10^−6^, respectively), with >80% of miRNAs putatively implicated (Table 3). This analysis suggests that these three biological pathways are highly involved in disease events.

## 4. Discussion

MiRNAs have emerged as an important class of biomarkers for detecting and monitoring various pathophysiological conditions present in all types of body fluids. In the present study, we found that miRNA detection in circulation did not discriminate JIA from SA well in children. Indeed, about 80% of the miRNAs in SE samples showed comparable expression in JIA and SA, and only 10% significantly differed between the two diseases. In contrast, in SF, >20% of detectable miRNAs significantly differed in expression in JIA versus SA, and almost half differed by at least two-fold in expression. Our study is unique in being the first to compare the SF in JIA and SA. We identify a specific miRNA based-signature, which was informative to define specific pathways involved in JIA. Such a signature may have potential as a diagnostic biomarker and/or for revealing novel targets for the development of treatments specific for JIA.

A signature of circulating miRNAs has been reported in SF from adult patients with rheumatoid arthritis (RA) and revealed that SF miRNAs have distinct patterns from plasma and joint miRNAs [25]. Murata et al. showed that plasma miRNAs can differentiate healthy donors from different types of adult rheumatisms, whereas SF miRNAs can discriminate different types of rheumatisms. Our study shows that depending on the biological fluid used, miRNA expression profiles differ, which highlights the different miRNA-mediated biological functions and cellular contexts. None of the serum miRNAs previously described in the literature as specific for JIA (such as miR-146a, miR-155, miR-132, miR-125a-5p, miR-26a, miR-16, and miR-223) are differently expressed in the serum of JIA compared to SA patients, and only 2 of them (miR-155 and miR-146a) are significantly different between the two forms of juvenile arthritis when investigating the SF. These data suggest that both diseases induce different molecular events, which are evidenced when investigating joints and not with blood.

Critical issues in miRNA profiling are non-uniform sample choice, handling, and processing, as well as cell contamination in sample preparation. Blood cells indeed contribute to the presence of miRNAs in circulation, markedly altering levels of specific miRNAs such as miR-451, miR-16, miR-92a, and miR-486 [26]. Perturbations in blood cell counts and hemolysis can also alter plasma miRNA levels by up to 50-fold. Here, we identified a 19-miRNA signature in SF that differentiates JIA and SA in children independently by sample centrifugation and is therefore not significantly contaminated by circulating cells. Importantly, because JIA involves a rare group of diseases, the cohort numbers were small but the 19-miRNA signature was still validated in two independent cohorts with the same technique. 

However, only 10 miRNAs were detectable in SF on RT-qPCR. This difference may be explained by various factors. First, HTG EdgeSeq is based on extraction-free chemistry, and RNA extraction for qPCR is known to induce biases; especially, miRNAs with low GC content can be lost during extraction [27,28]. In addition, depending on the extraction protocol and sample processing time, the miRNA profile may significantly differ. In a study of the reproducibility of plasma miRNA quantification for clinical studies, Rice et al. recommended that miRNAs be extracted from the plasma immediately after sampling or at least by <12 h and by using the miRneasy kit [29], which agrees with recommendations of the MIQE guidelines for miRNA quantification [21]. Second, because of high sensitivity of the HTG EdgeSeq WTA, the detection level is enhanced as compared with other technologies including qPCR, as reported in the miRQC report [29,30]. Thus, it is not surprising that some selected miRNAs are not detectable by qPCR. Moreover, because samples used for the discovery and validation cohorts were independent, both technical and biological variations can explain the non-confirmation of miRNAs but also show the robustness of the validated miRNAs as biomarker candidates.

Among the 19 miRNAs distinguishing the pathologies, >70% are new miRNAs listed in miRBase v20, and >50% have been previously found to be involved in the inflammatory response. Six miRNAs are implicated in autoimmune disorders, such as miR-342-5p in systemic lupus erythematosus or type 1 diabetes [31,32], or miR-150-5p [33], miR-146a-5p [34], miR-155-5p [35], miR-6716-5p [36], and miR-2909 [37,38] in RA. We found miR-146a and miR-155 significantly overexpressed in SF from children with JIA versus SA. MiR-146a is one of the most-studied miRNAs in RA and negatively regulates inflammation [39]. Its expression is increased in different cell types or tissues, including SF and synovial tissue from RA patients [35]. MiR-155, which is probably the second most described miRNA in RA, has also been found overexpressed in SF from RA patients [35]. It is mainly expressed by hematopoietic cells and positively regulates inflammation [40]. Recently, studies have identified several miRNAs, including miR-146a and miR-155, abnormally expressed in plasma or SE samples from patients with JIA as compared with healthy donors [13,14]. Together with these previous reports in JIA, our data suggest a common inflammatory component between juvenile and adult arthritis. To our knowledge, no functional analysis has been performed for miR-6764-5p. Further studies are needed to understand the implication of this new miRNA in inflammatory processes.

Seven of the 19 miRNAs identified were described in viral or bacterial infection responses. miR-342-5p suppresses coxsackievirus B3 biosynthesis [41], and miR-150 is deregulated in peripheral blood mononuclear cells of patients with sepsis versus healthy controls, with lower SE levels of miR-150 correlated with elevated sequential organ failure assessment (SOFA) scores and severity of sepsis [42]. Finally, miR-4667-5p is upregulated in response to *Burkholderia pseudomallei* infection in human lung epithelial cells [43]. 

The analysis of pathways regulated by the 19 miRNAs reveal key biological processes such as insulin, B-cell receptor, EGF/EGFR signaling, and pathways involving cardiomyocytes. Systemic abnormalities in growth hormone/insulin-like growth factor 1 and B-cell receptor signaling are known to be involved in JIA [44,45]. Among the JIA subtype-specific blood gene expression profiles identified by Barnes et al., interleukin 2, B-cell receptor and JAK/STAT signaling pathways were over-represented in persistent oligoarthritis [46]. Surprisingly, we found two pathways involving heart cells such as calcium regulation in cardiac cells and cardiomyocyte hypertrophy, which agrees with previous publications showing cardiac involvement in systemic JIA [47]. Although not described yet in JIA, the prevalence of insulin resistance is higher in RA patients than healthy individuals [48]. Overall, data from the literature suggest that the 19 miRNAs differentiating JIA and SA in children are associated with autoimmune versus infectious mechanisms, which is relevant to the respective etiology of both diseases.

Even though the differential diagnosis between SA and JIA in children is critical for proper care and prevention of joint lesions, we have no reliable biomarkers to distinguish between these two forms of arthritis at symptom onset. Previously, Aupiais et al. retrospectively compared clinical and biological features of SA and JIA but did not find any specific blood markers differentiating the diseases [49]. The authors concluded that even if according to standard guidelines, children with joint effusion associated with fever, high CRP level, and WBC count are suspected to have SA and should undergo joint aspiration and antibiotic treatments; this is insufficient to discriminate JIA and SA. Although CRP level has been described as a predictor of SA [50], our cohorts showed an inconsistent significance of the factor, which might be related to the infectious agent (*K. kingae*), as previously suggested [18]. We also observed a significant difference between WBC count in JIA and SA, but in 4 of 9 children with SA, WBC counts were comparable to those in JIA (Table 2). To our knowledge, our study is the first prospective study to compare SF in both diseases and successfully identifiy a combination of three miRNAs (miR-6764-5p, miR-155, and miR-146a-5p) with high specific and sensitive diagnostic potential in JIA (AUC = 1 and *p* = 0.0017). 

## 5. Conclusions

Our study is the first to investigate the SF miRNome in JIA, especially in comparison with SA in children. Our findings identified a 19-miRNA signature in SF in JIA; in particular, a highly sensitive and specific three-miRNA signature (miR-146a-5p, miR-155, and miR-6764-5p) that could be a useful biomarker of disease and a novel track to study biological functions strongly involved in JIA pathophysiology. Because joint punctures are an integral part of the management of SA, our data also suggest that measuring the expression of these three miRNAs upon initial presentation might be an easy and rapid step for rapid initiation of optimal treatment and care management. Although our study is encouraging, future prospective studies with an increased number of patients should be conducted as well as an analysis of the expression of key genes of the identified pathogenic pathways.

## Figures and Tables

**Figure 1 cells-08-01521-f001:**
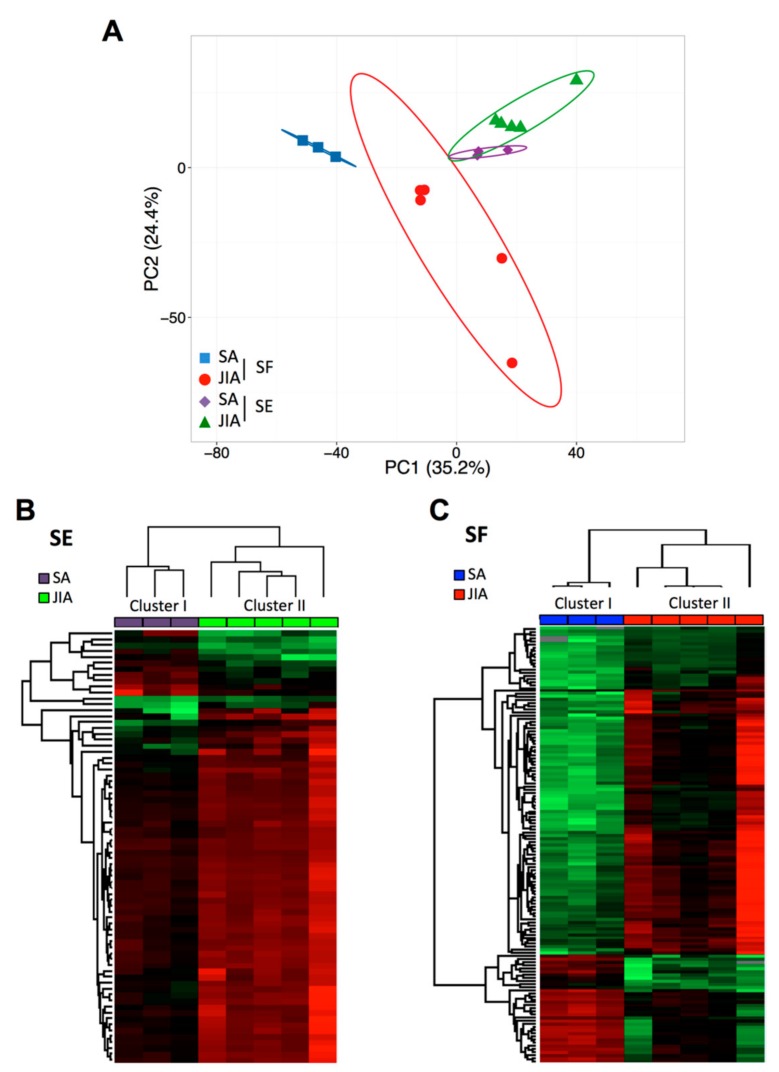
Principal component analysis (PCA) and cluster analysis of miRNA expression in articular synovial fluid (SF) and serum (SE) samples from children with juvenile idiopathic arthritis (JIA) and septic arthritis (SA). (**A**) Global PCA of miRNA expression between SF (red and blue) and SE (violet and green) samples. (**B**) Supervised hierarchical clustering analysis of miRNA expression profile in SE samples from children with JIA (n = 5; green) and SA (n = 3; violet). (**C**) Supervised hierarchical clustering analysis of miRNA expression profile in SF samples from children with JIA (n = 5; red) and SA (n = 3; blue). Columns represent sample type and rows miRNAs. A color map is used to detect differences in expression: green indicates downregulation, red indicates upregulation.

**Figure 2 cells-08-01521-f002:**
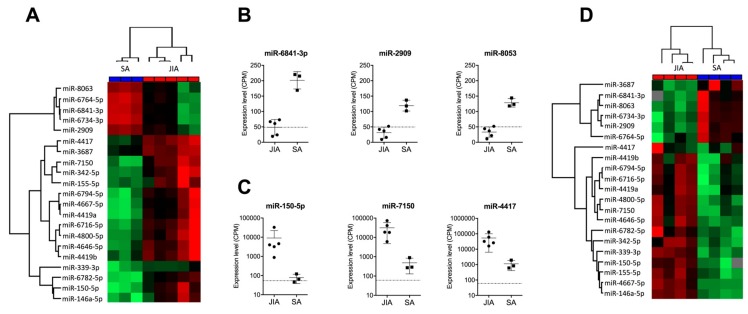
Identification of a miRNA-based signature in SF discriminating JIA from SA. (**A**) Hierarchical clustering of the 21 miRNAs deregulated in SF, selected with *p* < 0.01, fold change in expression >4 or <–2 between JIA and SA, and >100 CPM expression. A color map is used to detect differences in expression: green indicates downregulation; red indicates upregulation. (**B**) Expression of the three most downregulated miRNAs in SF from children with JIA versus SA. (**C**) Expression of the three most upregulated miRNAs in SF from children with JIA versus SA. (**D**) Hierarchical clustering of the 21 miRNAs detected in SF from a new prospective cohort of non-centrifuged samples.

**Figure 3 cells-08-01521-f003:**
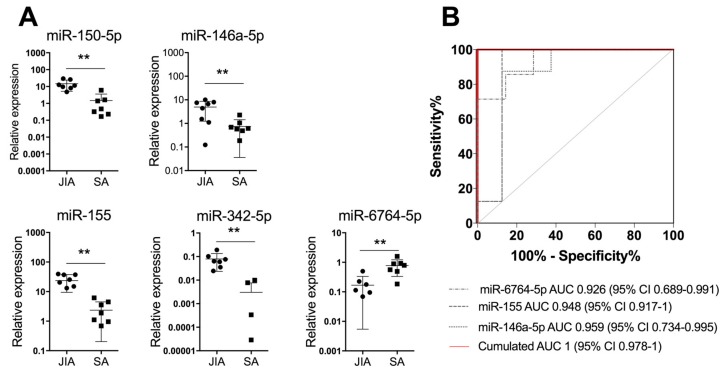
A combination of three miRNAs in SF differentiates JIA from SA at initial presentation. (**A**) Scatterplots show relative expression levels of miR-150-5p, miR-146a-5p, miR-155, miR-342-5p, and miR-6764-5p in SF from children with JIA (n = 9) and SA (n = 9). ** *p* < 0.01 by non-parametric Mann–Whitney test. (**B**) Area under the receiver operating characteristic (ROC) curve (AUC) values were estimated by using mROC software. ROC curves for miRNAs with statistically significant results and their reference lines are provided. The best linear combination (virtual marker) of miRNA expression is maximized to discriminate JIA and SA (red line): AUC = 1 and *p* = 0.0017.

**Table 1 cells-08-01521-t001:** Clinical and biological characteristics of synovial fluid samples from children with juvenile idiopathic arthritis (JIA) and septic arthritis (SA) in the exploration and validation cohorts (centrifuged or not).

	Exploration Cohort			Validation Cohort					
	Centrifuged			Non-Centrifuged			Centrifuged		
	JIA (n = 5)	SA (n = 3)	*p* Value	JIA (n = 4)	SA (n = 4)	*p* Value	JIA (n = 9)	SA (n = 9)	*p* Value
**Age at sampling (years)**	10.3 ± 3.7	1.2 ± 0.1	0.036	7.5 ± 3.9	1.7 ± 0.6	0.057	8.9 ± 3.7	1.2 ± 0.2	0.0002
**Number of affected joints % (n)**									
**Monoarthritis**	40 (2)	100 (3)	/	50 (2)	100 (4)	/	55.5 (5)	100 (9)	/
**Oligoarthritis**	60 (3)	0 (0)	/	50 (2)	0 (0)	/	45.5 (4)	0 (0)	/
**Affected joints, % (n)**									
**Knee**	100 (5)	100 (3)	/	4	1	/	100 (9)	100 (9)	/
**Hip**	0 (0)		/	0	2	/	0 (0)	0 (0)	/
**Ankle**	40 (2)		/	0	0	/	22.2 (2)	0 (0)	/
**Shoulder**	20 (1)		/	0	1	/	0 (0)	0 (0)	/
**Bacterial contamination, % (n)**									
***Kingella kingae***	NA	100 (3)	/	NA	100 (4)	/	NA	100 (9)	/
**C- reactive protein level**	20.3 ± 33.5	23.9 ± 4.4	ns	5.3	48.0 ± 24.6	NA	13.6 ± 23.0	28.6 ± 15.6	0.042
**Blood WBC count (×10^9^/L)**	6.8 ± 1.1	12.7 ± 1.8	0.018	4.4	10.1 ± 4.3	NA	7.1 ± 2.9	12.3 ± 3.6	0.002
**Neutrophil count (×10^9^/L)**	3.8 ± 1.0	7.5 ± 1.6	ns	NA	5.5 ± 3.6	NA	4.0 ± 2.1	4.4 ± 3.0	ns
**Haemoglobin level (g/dL)**	11.9 ± 1.0	11.4 ± 0.9	ns	NA	11.1 ± 1	NA	11.9 ± 4.0	11.4 ± 4.0	ns
**Platelet count (×10^9^/L)**	310 ± 65	368 ± 29	ns	NA	501 ± 176	NA	320 ± 119	385 ± 144	ns

NA: not applicable; ns: not significant; WBC: whole blood cell.

**Table 2 cells-08-01521-t002:** Selected miRNAs differentially regulated in synovial fluid samples (not centrifuged) from children with JIA versus SA.

	Centifuged		Non-Centrifuged	
miRNA Name	log_2_ (FC)	*p*-Values	log_2_ (FC)	*p-Values*
miR-146a-5p	5.08	0.010	6.90	*8.8 × 10^−6^*
miR-150-5p	5.95	0.002	6.08	*0.0008*
miR-155-5p	4.00	0.022	5.22	*0.0007*
miR-2909	−2.19	0.01	−2.12	*0.002*
miR-339-3p	4.32	0.006	7.54	*0.00008*
miR-342-5p	4.57	0.004	2.25	*0.001*
miR-3687	4.42	0.0007	−1.13	*0.012*
miR-4417	5.34	0.0005	0.42	*0.395*
miR-4419a	4.21	0.002	4.20	*0.020*
miR-4419b	4.19	0.005	1.68	*0.105*
miR-4646-5p	4.21	0.004	1.75	*0.004*
miR-4667-5p	4.01	0.005	3.29	*0.00003*
miR-4800-5p	4.83	0.005	2.83	*0.003*
miR-6716-5p	4.89	0.002	4.86	*0.009*
miR-6734-3p	−2.09	0.005	−1.58	*0.003*
miR-6764-5p	−2.12	0.006	−1.75	*0.022*
miR-6782-5p	4.76	0.0003	4.18	*0.005*
miR-6794-5p	4.16	0.004	3.37	*0.010*
miR-6841-3p	−2.24	0.007	−1.38	*0.022*
miR-7150	5.79	0.0007	4.90	*0.004*
miR-8063	−2.14	0.009	−2.37	*0.002*

Expression of miRNAs is plotted as fold change (FC) in SF samples from children with JIA (n = 5) versus SA (n = 3).

**Table 3 cells-08-01521-t003:** Biologic pathways enriched by differentially expressed miRNAs in children with JIA and SA.

Path Name	No. of mRNAs	miRNA Names	FDR *p*-Value
**Insulin signaling**	18	miR-146a-5p, miR-150-5p, miR-155-5p, miR-2909, miR-339-3p, miR-342-5p, miR-4419a, miR-4419b, miR-4646-5p, miR-4667-5p, miR-6716-5p, miR-6734-3p, miR-6764-5p, miR-6782-5p, miR-6794-5p, miR-7150, miR-6841-5p, miR-8063	3.4 × 10^−2^ to 1.3 × 10^−7^
**B-cell receptor signaling**	16	miR-146a-5p, miR-150-5p, miR-155-5p, miR-2909, miR-339-3p, miR-342-5p, miR-4419a, miR-4419b, miR-4646-5p, miR-4667-5p, miR-4800-5p, miR-6734-3p, miR-6782-5p, miR-6794-5p, miR-7150, miR-8063	4.8 × 10^−2^ to 6.5 × 10^−6^
**EGF EGFR signaling**	16	miR-146a-5p, miR-150-5p, miR-155-5p, miR-2909, miR-339-3p, miR-342-5p, miR-4419a, miR-4419b, miR-4667-5p, miR-4800-5p, miR-6716-5p, miR-6734-3p, miR-6782-5p, miR-6794-5p, miR-7150, miR-8063	4.5 × 10^−2^ to 2.4 × 10^−6^
**Calcium regulation in cardiac cell**	15	miR-150-5p, miR-2909, miR-342-5p, miR-4419a, miR-4419b, miR-4646-5p, miR-4667-5p, miR-4800-5p, miR-6716-5p, miR-6734-3p, miR-6764-5p, miR-6782-5p, miR-6794-5p, miR-7150, miR-8063	4.5 × 10^−2^ to 7.9 × 10^−8^
**DNA damage response only ATM-dependent**	15	miR-146a-5p, miR-150-5p, miR-155-5p, miR-2909, miR-339-3p, miR-342-5p, miR-4419a, miR-4419b, miR-4646-5p, miR-4667-5p, miR-6716-5p, miR-6782-5p, miR-6794-5p, miR-7150, miR-8063	4.9 × 10^−2^ to 8.1 × 10^−5^
**MiRNAs in cardiomyocyte hypertrophy**	15	miR-146a-5p, miR-150-5p, miR-155-5p, miR-2909, miR-342-5p, miR-4419a, miR-4419b, miR-4646-5p, miR-4667-5p, miR-4800-5p, miR-6782-5p, miR-6794-5p, miR-7150, miR-6841-5p, miR-8063	3.2 × 10^−2^ to 7.9 × 10^−7^
**Adipogenesis**	13	miR-146a-5p, miR-150-5p, miR-2909, miR-339-3p, miR-342-5p, miR-4419a, miR-4419b, miR-4800-5p, miR-6734-3p, miR-6764-5p, miR-6782-5p, miR-6794-5p, miR-7150, miR-8063	4.6 × 10^−2^ to 5.6 × 10^−5^
**G-protein signaling**	13	miR-146a-5p, miR-150-5p, miR-2909, miR-339-3p, miR-342-5p, miR-4419a, miR-4419b, miR-4667-5p, miR-4800-5p, miR-6734-3p, miR-6764-5p, miR-6782-5p, miR-7150, miR-8063	4.7 × 10^−2^ to 1 × 10^−4^
**Regulation of actin cytoskeleton**	12	miR-146a-5p, miR-155-5p, miR-2909, miR-342-5p, miR-4419a, miR-4419b, miR-4646-5p, miR-4667-5p, miR-6782-5p, miR-6794-5p, miR-7150, miR-6841-5p, miR-8063	4.1 × 10^−2^ to 8.3 × 10^−6^
**MAPK signaling**	12	miR-146a-5p, miR-155-5p, miR-339-3p, miR-342-5p, miR-4419a, miR-4646-5p, miR-4667-5p, miR-6734-3p, miR-6794-5p, miR-7150, miR-6841-5p, miR-8063	4.8 × 10^−2^ to 5 × 10^−5^
**Myometrial relaxation and contraction**	12	miR-150-5p, miR-2909, miR-342-5p, miR-4419a, miR-4667-5p, miR-4800-5p, miR-6716-5p, miR-6734-3p, miR-6764-5p, miR-6782-5p, miR-7150	4.7 × 10^−2^ to 6 × 10^−4^
**TGF beta signaling pathway 1**	12	miR-146a-5p, miR-155-5p, miR-2909, miR-342-5p, miR-4419a, miR-4419b, miR-4646-5p, miR-4667-5p, miR-6734-3p, miR-7150, miR-6841-5p, miR-8063	4.8 × 10^−2^ to 7.6 × 10^−9^
**AMPK signaling**	11	miR-2909, miR-342-5p, miR-4419a, miR-4419b, miR-4646-5p, miR-6716-5p, miR-6734-3p, miR-6764-5p, miR-6782-5p, miR-6794-5p, miR-8063	4 × 10^−2^ to 8 × 10^−4^
**Leptin signaling**	11	miR-146a-5p, miR-2909, miR-342-5p, miR-4419b, miR-4646-5p, miR-4667-5p, miR-6734-3p, miR-6782-5p, miR-6794-5p, miR-7150, miR-8063	4.8 × 10^−2^ to 2 × 10^−4^
**Apoptosis**	9	miR-146a-5p, miR-150-5p, miR-155-5p, miR-2909, miR-339-3p, miR-342-5p, miR-4419b, miR-6794-5p, miR-8063	2.7 × 10^−2^ to 4 × 10^−4^
**RANKL RANK signaling**	9	miR-150-5p, miR-155-5p, miR-2909, miR-339-3p, miR-342-5p, miR-3687, miR-4667-5p, miR-6794-5p, miR-6841-5p	4.8 × 10^−2^ to 5 × 10^−4^
**ErbB signaling**	8	miR-150-5p, miR-2909, miR-342-5p, miR-4667-5p, miR-6734-3p, miR-6782-5p, miR-6794-5p, miR-7150	4.8 × 10^−2^ to 4 × 10^−4^
**Nuclear receptors**	8	miR-150-5p, miR-155-5p, miR-342-5p, miR-4419a, miR-4667-5p, miR-6734-3p, miR-6782-5p, miR-6794-5p, miR-7150	4.8 × 10^−2^ to 3.5 × 10^−5^
**Prolactin signaling**	8	miR-2909, miR-342-5p, miR-4419a, miR-4646-5p, miR-4667-5p, miR-6716-5p, miR-6734-3p	4.6 × 10^−2^ to 1 × 10^−4^

FDR, false discovery rate; EGF, epidermal growth factor; EGFR, EGF receptor; ATM, ataxia telangiectasia mutated; MAPK, mitogen-activated protein kinase; TGF, transforming growth factor; AMPK, 5’ AMP-activated protein kinase; RANKL, receptor activator of nuclear factor kappa-Β ligand; ErbB, Erb-B2 receptor tyrosine kinase.

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
