# Peer review of "Synovial-Fluid miRNA Signature for Diagnosis of Juvenile Idiopathic Arthritis"

_cells, 2019, doi:10.3390/cells8121521_

Round 1

Reviewer 1 Report

The method of library preparation for sequencing (EdgeSeq) is very unusual compared to standard small RNA sequencing protocols. Its advantages are unclear, while potential disadvantages –biases introduced by the protection probes – are obvious. The authors should justify the use of this method by comparison to established protocols. Indeed, looking at the list of “miRNAs”, many of the discussed molecules are probably NOT miRNAs. For example, miR-4417 is ribosomal RNA as miRBase itself acknowledges. Most likely, almost all miRNA numbered > 1300 are NOT bona fide The title should state the control disease; K. kingae septic arthritis. Line 89: “lumbar puncture”? Along the results, it is not always clear when are the exploratory results and when validation results are presented. Line 135 1st word is misspelled. Line 169: how did you log-transform 0 values? Line 173: for qPCR? if so, make clear in section heading Table 1: p-values are missing and replaced with “/” symbols Figure 1: which miRNA are included – obviously not all are included. Line 223: there is certainly no “clear separation” Table 2: which cohort? Line 248-249: it is not clear how the authors determined qPCR detection limit. Also, the claim that qPCR is less sensitive than EdgeSeq (line 340) lacks any support and is unlikely to be valid. Figure 2: “miR-7150” and “miR-4417” seem to be the highest expressed miRNA. Do the authors really believe they are miRNA? If these are truly miRNA with such high expression, why was their discovery so late!? Line 263: what is the “2nd exploratory cohort”? Discussion: the sentence “suggests that a large number of the associated signaling pathways might be identical” is ridiculous – the miRNA profiles are similar because the inflammatory joint processes have very little effect upon the miRNA in the circulation.

Reviewer 2 Report

Many studies have assessed miRNAs as possible biomarkers for arthritis. Nevertheless in several studies miRNA levels and subtypes don't allow to distinguish specific pattern in order to fulfill specific diagnostic criteria for different arthropaties.

The manuscript submitted for the first time demonstrates the presence of a specific miRNA signature in synovial fluid of patients affected from JIA and doing so, lays the groundwork for a diagnostic use of this marker in synovial fluid.

The experimental design is adequate and follow the standard and the requirements of other outstanding studied about synovial fluid and miRNAs; methodology is described very accurately and I share completely the conclusions as are described and discussed in the submitted paper.

Figures, bibliography and tables are clear and complete.

There is only one issue that I wanted more developed and discussed and this is about the biological functions of specific miRNA described however this probably lies outside the framework of this paper.

For these reasons i suggest to editors to accep the manuscrit submitted without any modfications.

Author Response

We are thankful to Cells’ editor and all 5 reviewers for their in-depth evaluation and comments that will help improving our manuscript. Thank you for this inspiring comment.

Reviewer 3 Report

In this manuscript authors have beautifully identified the role of synovial fluid micro RNA levels in diagnosis of the juvenile idiopathic arthritis. In the study authors have used next gen sequencing to identify the signature of the micro RNA signature as a biomarker for the diagnosis of juvenile idiopathic arthritis. The study is indeed an interesting one and the conclusion supports the study.

Specific comments:

Authors have used next gen sequencing to identify the micro RNA signature as a diagnostic tool for juvenile idiopathic arthritis. While analysis the sequencing data have the authors noted any novel microRNA signatures in addition to the novel ones? Authors have used both centrifuged and non centrifuged samples to identify miRNA levels what the clinical significance in using centrifuged and non centrifuged one?

Reviewer 4 Report

Nziza et al. suggest that SF miRNAs could be promising biomarkers to identify JIA & SA patients. Although the experiments are well designed, and manuscript is enthusiastic, I have following comments:

Firstly, already published reports indicate that certain miRNA markers in SE (for eg. miRNAs 146a, 155, 132, etc) are important for identification and in prognosis of the JIA patients. And SE is relatively convenient method to obtain from the children than SF (which is an invasive technique). Do authors suggest to implicate the importance of rather difficult and invasive technique to isolate SF for the identification of JIA cases rather than drawing the blood? Or the results shows in this manuscript are just “Informative” that SF has also certain miRNA biomarkers signature? Secondly, authors show that they were unable to identify the differences in miRNA expression between JIA and SA children’s SE samples (table s1 in the paper). Again, was not it feasible to confirm if the result by including healthy SE samples, considering the authors concern that they were unable to obtain SF from healthy children (Line 70-76), but obtaining SE could have been feasible and also could be informative if assays are working itself. Because, as mentioned in comment a., it would have been a validation experiment to see if already well established markers in SE are replicable in this manuscript. This result could have provided authors with the idea that experimental results are valid, and their observation of not finding any difference in the miRNAs between SA and JIA is not false negative.

Reviewer 5 Report

Dear authors,

To my opinion this is an excellent manuscript, presenting relevant and novel information that is valuable for future diagnostics. It is very well written, clear, scientifically sound, with a well rounded discussion of results. 

I would only like to give some 'food for thought': In rheumatoid arthritis, it appears that the induction of cross-reactive ('autoimmune' is likely only half of the recognition) antibodies to certain opportunistic pathogens is somewhere at the beginning of pathogenesis (especially studies by prof. Alan Ebringer on Proteus Mirabilis are recommended) in genetically susceptible hosts, especially relating to shared epitopes in HLA molecules. It appears that the induction site is often mucosal surfaces such as lungs, oral cavity, GI-tract or urinary tract. It could be worth while to combine genetic studies of JIA patients with microbial screening, or at least testing for intestinal barrier function / a questionnaire for history of gingivitis or UTI infections for example, and to see if there are correlations with the miRs that were discovered. 

Round 2

Reviewer 1 Report

Most of the discussed molecules are NOT miRNA - they may exist in the fluid, and may even be abundant, but they are not miRNA. miRBase itself include a note on almost all of these miRNA:

"Annotation confidence: not enough data"

Reviewer 4 Report

n.a